# Myocardial infarction and stroke subsequent to urinary tract infection (MISSOURI): protocol for a self-controlled case series using linked electronic health records

Nicola F Reeve ,[1] Victoria Best,[2] David Gillespie ,[3] Kathryn Hughes,[4] Fiona V Lugg-Widger ,[3] Rebecca Cannings-John ,[3] Fatemeh Torabi,[2] Mandy Wootton,[5] Ashley Akbari ,[2] Haroon Ahmed[1]

For numbered affiliations see end of article.

**Correspondence to**
Dr Nicola F Reeve;
reeven1@cardiff.ac.uk

## ABSTRACT

**Introduction** There is increasing interest in the relationship between acute infections and acute cardiovascular events. Most previous research has focused on understanding whether the risk of acute cardiovascular events increases following a respiratory tract infection. The relationship between urinary tract infections (UTIs) and acute cardiovascular events is less well studied. Therefore, the aim of this study is to determine whether there is a causal relationship between UTI and acute myocardial infarction (MI) or stroke.

**Methods and analysis** We will undertake a self-controlled case series study using linked anonymised general practice, hospital admission and microbiology data held within the Secure Anonymised Information Linkage (SAIL) Databank. Self-controlled case series is a relatively novel study design where individuals act as their own controls, thereby inherently controlling for time-invariant confounders. Only individuals who experience an exposure and outcome of interest are included.

We will identify individuals in the SAIL Databank who have a hospital admission record for acute MI or stroke during the study period of 2010–2020. Individuals will need to be aged 30–100 during the study period and be Welsh residents for inclusion. UTI will be identified using general practice, microbiology and hospital admissions data. We will calculate the incidence of MI and stroke in predefined risk periods following an UTI and in 'baseline' periods (without UTI exposure) and use conditional Poisson regression models to derive incidence rate ratios.

**Ethics and dissemination** Data access, research permissions and approvals have been obtained from the SAIL independent Information Governance Review Panel, project number 0972. Findings will be disseminated through conferences, blogs, social media threads and peer-reviewed journals. Results will be of interest internationally to primary and secondary care clinicians who manage UTIs and may inform future clinical trials of preventative therapy.

## STRENGTHS AND LIMITATIONS OF THIS STUDY

⇒ The self-controlled case series method controls for time-invariant confounding, enabling more reliable causal estimates of the association between urinary tract infection (UTI) and acute myocardial infarction (MI) or stroke, compared with between-individual study designs.

⇒ A causal relationship between UTI and acute MI or stroke has implications for our understanding of cardiovascular disease mechanisms and may inform new methods of disease prevention.

⇒ Using individual-level population-scale anonymised, routinely collected electronic health record (EHR) data provides adequate power to study subgroups and maximises representativeness and generalisability.

⇒ EHR data are collected and recorded for clinical purposes, and therefore, the reliability of research findings is dependent on the quality and completeness of these data.

⇒ Clinical and microbiological diagnoses of UTI are subject to caveats, and therefore, we will use several definitions of UTI that use the different data sources in the Secure Anonymised Information Linkage Databank.

## INTRODUCTION

Since the late 1990s, an increasing number of observational studies have found an association between acute infections and myocardial infarction (MI).[1–10] Most studies focused on respiratory tract infections (RTIs), and found an increased risk of acute MI in the 1–3 days following an RTI, with the effect size varying according to the infecting organism.[2–8] For example, Kwong *et al* found a sixfold increase in the risk of MI in the week after influenza infection, a fourfold increase after respiratory syncytial virus and a threefold increase after

**Table 1** Model assumptions and solutions to violations of those assumptions

| Assumption | How the assumption applies to this study | Solution | Example of use of the solution in the literature |
|---|---|---|---|
| Subsequent exposures should not be affected by previous events. | We might see a temporary increase in UTIs subsequent to an MI or stroke event, which would bias estimates towards the null. | Apply a prerisk period. | Gibson et al studied the association between prescription drugs and road traffic accidents. As some drugs may be used to treat anxiety or pain caused by the crash, a 4-week pre-exposure period was included.[37] |
| | As both MI and stroke have relatively high death rates, the length of the observation period is dependent on events, and no further exposures are possible after death. | Use an event-dependent observation period model extension[38] and conduct a sensitivity analysis that repeats the analysis, excluding individuals who died within 30 days of the event. | Bruer et al used the event-dependent observation period model extension in their study on the association between antipsychotic drugs and myocardial infarction.[39] Langan et al studied the risk of stroke following herpes zoster. They conducted a sensitivity analysis excluding individuals who died within 90 days of stroke.[40] |
| Event rates are constant within defined periods | MI and stroke are more common in older individuals and may be affected by seasonal changes. | Control for age and season effects. | Grave et al studied the association between seasonal influenza vaccination and Guillain-Barré syndrome. They adjusted for calendar month, as the vaccinations are seasonal by design.[41] In a study of the association between chickenpox and stroke, Thomas et al adjusted for age in 5-year age bands.[42] |
| Events are independently recurrent or rare. | MI and stroke are not independent: once an individual has a first event, they are more likely to have a second. | Study first events only. | Langan et al began the observation period 12 months into follow-up time to ensure first stroke events had been correctly identified.[40] |

MI, myocardial infarction; UTIs, urinary tract infections.

other respiratory viruses.[5] Several studies have also found evidence of an association between pneumonia and acute cardiovascular events (including MI and stroke).[9–13] The increased risk of acute cardiovascular events following pneumonia infection persists for up to 10 years.[9] This long-term risk of acute cardiovascular events has also been observed after other severe infections, including sepsis and bacteraemia.[14–17]

It is thought that acute infection may cause major cardiovascular events through three mechanisms. First, the inflammatory response from acute infection may destabilise atherosclerotic plaques. Second, the prothrombotic, procoagulant state associated with acute infection may increase the risk of thrombosis at the site of plaque disruption. Third, inflammation and fever lead to an increase in heart rate, which may cause 'demand ischaemia' if the metabolic demands of the myocardial cells exceed oxygen supply.[1]

Urinary tract infections (UTIs) can affect any part of the urinary system, including the kidneys, ureters, bladder and urethra. Most infections involve the lower urinary tract: the bladder and the urethra. UTIs are common infections, with 37% of women reporting experiencing at least one in their lifetime, and 29% experiencing more than one.[18] UTIs are associated with considerable morbidity. The global burden of disease study 2010 estimates the disability-adjusted life-years attributable to tubulointerstitial nephritis, pyelonephritis and UTI to be 45 (95% uncertainty interval 32–55) per 100 000 population.[19]

The relationship between UTIs and acute cardiovascular events is less well studied than for RTIs. Only one previous study has examined this relationship. Smeeth et al used the self-controlled case series (SCCS) method to analyse data from the General Practice Research Database.

They found increased rates of MI and stroke subsequent to UTI, with the risk being highest in the first 3 days.[6] However, the data analysed are almost 20 years old, and there have been no attempts to replicate the findings. Furthermore, the study defined UTI using clinical codes only, so it is unclear if the reported associations related to individuals with clinical symptoms alone, or symptoms and bacteriuria, making it difficult to interpret whether individuals had true UTIs, or whether non-specific symptoms were misdiagnosed as UTI but represented early signs of a cardiovascular event. In addition, other studies have found that roughly two-thirds of women suspected to have UTI on presentation to primary care have no evidence of UTI on microbiological culture.[20] Therefore, the use of clinical codes alone to define UTI may lead to bias from misclassification of the exposure.

Therefore, the aim of this study is to determine whether there is a causal relationship between UTI and acute MI or stroke by analysing linked general practice, hospital admission and microbiology data, from a representative sample of the Welsh population. We will use the SCCS method, which controls for time-invariant confounding, enabling us to more reliably draw causal inferences between UTI and acute MI or stroke, compared with between-individual study designs.

## METHODS AND ANALYSIS
### Aims and objectives
The specific objectives of this research are to:
1. Estimate incidence rate ratios (IRRs) for acute MI and stroke in the 90 days following a clinically suspected and microbiologically confirmed UTI compared with baseline (all times outside of the 90-day risk period).

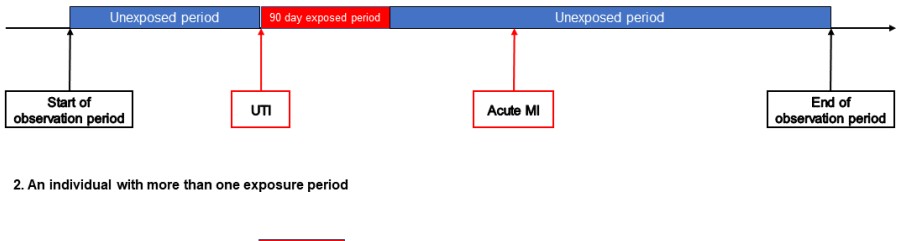

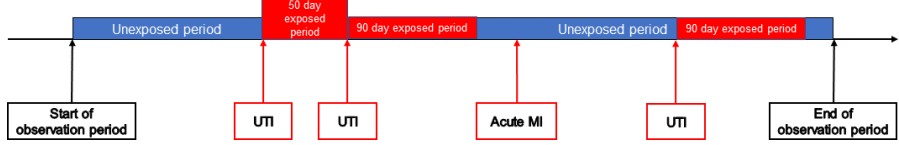

**Figure 1** Diagrammatic representation of observation time for an individual in the proposed self-controlled case series design. MI, myocardial infarction; UTI, urinary tract infection.

2. Assess the effect of different methods of UTI ascertainment on estimated rate ratios (ie, clinically suspected and microbiologically confirmed; clinically diagnosed only; clinically suspected but not supported by microbiology and UTI diagnosed and treated in hospital).
3. Investigate whether the effect of a clinically suspected and microbiologically confirmed UTI on acute MI and stroke differs according to the infecting organism.

Our primary hypothesis is that a clinically suspected and microbiologically confirmed UTI will increase the risk of acute MI or stroke in the 0–90 days postinfection period.

### Data

We will use the Secure Anonymised Information Linkage (SAIL) Databank. This is an internationally recognised trusted research environment (TRE), with robust secure storage, enabling access to anonymised, linkable, individual-level Welsh population-scale data for research, with a focus on improving population health and health services. Data within SAIL are pseudonymised and made available to approved projects and users following an application to, and approval from the independent information governance review panel (IGRP). SAIL's storage and linkage processes ensure anonymity: first, data sources being provided to SAIL are split as per the standard split file process, with the source organisation splitting the source data into demographic data and clinical data, with a system linkage field to allow data to be rejoined later. This addresses confidentiality and disclosure issues that arise when working with health data by separating easily recognised person-based variables such as name and date of birth from clinical data, including information on diagnoses, tests and prescriptions. The demographic data are anonymised and assigned an anonymised linkage field. These split files are then joined together using the system linkage field by the SAIL team and made available to researchers following encryption.[21–23]

We will use the SAIL Databank to access the following linked data: Welsh Longitudinal General Practice data (WLGP), Patient Episode Database for Wales (PEDW),

Welsh Results Reporting Service (WRRS). The WLGP contains data from 84% of general practices in Wales, consisting of longitudinal data for 2.6 million Welsh residents, representing 84% of the population.[24] Demographic data, clinical diagnoses and prescription data are included. The PEDW contains International Classification of Disease version 10 (ICD-10) coded diagnoses for admissions to any Welsh hospital.[25] The WRRS contains data on all tests requested from primary and secondary care National Health Service (NHS) Wales organisations processed and analysed in NHS Wales laboratories, including requests for urine microscopy, culture and antibiotic susceptibilities.[26] Data are available from the data sources at varying times based on when clinical information systems began, with data quality improving over time. As such, for our study, based on our approvals and where the data sources have consistent coverage and quality, we will be using them between 1 January 2010 and 31 December 2020.

### Study design

Individuals who experience UTIs and individuals who do not can differ in unmeasured ways and hence could be sources of residual confounding. We will use the SCCS design method to deal with this issue. The SCCS method is an epidemiological study design for which individuals act as their own control so that both measured and unmeasured characteristics that vary between individuals are completely controlled.[27] Only individuals who have experienced an outcome and exposure of interest are included. The SCCS method compares the incidence of an outcome during predefined risk periods with incidence during baseline periods (all times outside of risk and prerisk periods) and estimates the temporal association between a transient exposure and outcome. Time-invariant covariates (eg, sex) are inherently controlled for, and time varying covariates (eg, age) are adjusted for within the models. The method was originally developed to investigate associations between vaccinations and acute adverse events, such as aseptic meningitis,[28 29] but has since been applied in a range of epidemiological

**Table 2** Definitions of UTI for primary and secondary analyses. As the UTI definitions are combinations of two or more components, the start of the risk period is defined as the date of the earliest component

| | UTI-related read code in GP data (online supplemental material B) | Antibiotic prescription in GP data (online supplemental material C) | UTI-related ICD-10 code in PEDW (online supplemental material D) | Urine culture results in WRRS | Time frame | Clinical scenario |
|---|---|---|---|---|---|---|
| Primary analysis | Yes | Yes | No | Yes, showing bacterial growth of ≥1x10⁸CFU/L and WCC ≥1x10⁸/L | Three codes occur within a 7-day window | GP clinically suspected and microbiologically confirmed UTI |
| Secondary analysis 1 | Yes | Yes | No | Yes, showing mixed bacterial growth (any descriptor for 'mixed growth' or >3 organisms). | Three codes occur within a 7-day window | GP clinically suspected UTI with mixed growth |
| Secondary analysis 2 | Yes | Yes | No | No | Same day | GP clinically diagnosed and treated UTI. It is important to consider this group as not all individuals with suspected UTI have urine culture, and limiting to those with culture is subject to selection bias. |
| Secondary analysis 3 | Yes | Yes | No | Yes, showing bacterial growth of <1x10⁷CFU/L. | Three codes occur within a 7-day window | UTI is clinically suspected but not supported by microbiology. This group is important to understand if early symptoms and signs of acute MI or stroke are attributed to UTI. |
| Secondary analysis 4 | No | No | Yes | Yes, showing bacterial growth of ≥1x10⁸CFU/L and WCC ≥1x10⁸/L | Two codes occur within a 7-day window | UTI diagnosed and/or treated in hospital |
| Secondary analysis 5 | Yes, OR ICD-10 code | Yes, OR ICD-10 code | Yes, OR: UTI Read code AND antibiotic Read code | Yes, showing bacterial growth of ≥1x10⁸CFU/L and WCC ≥1x10⁸/L | Two/three codes occur within a 7-day window | GP clinically suspected and microbiologically confirmed UTI or UTI diagnosed and/or treated in hospital |

GP, General Practice; ICD-10, International Classification of Disease version 10; PEDW, Patient Episode Database for Wales; UTI, urinary tract infection; WCC, white cell count; WRRS, Welsh Results Reporting Service .

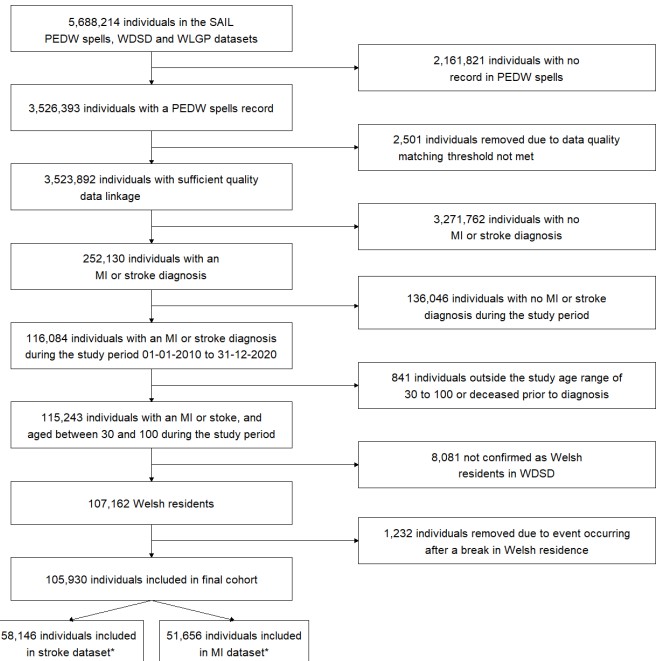

**Figure 2** Case selection process. *Individuals appear and are counted in both the stroke and MI datasets if they had both an MI and a stroke event within the study period. MI, myocardial infarction; PEDW, Patient Episode Database for Wales; SAIL, Secure Anonymised Information Linkage; WLGP, Welsh Longitudinal General Practice data; WDSD, Welsh Deographic Service Dataset.

settings,[30–32] including non-acute events such as autism.[33] As with other study designs, the SCCS method makes several assumptions that need to be met in order to obtain valid and unbiased estimates, but there are model extensions which provide solutions to violations of these assumptions under certain circumstances.[34] The model assumptions, how they apply to our study, and the solutions to violations of the assumptions are given in table 1.

A diagrammatic representation of observation time for an individual in the proposed SCCS design is given in figure 1. Risk periods start on the date of an UTI and end 90 days later. The date of an UTI is the earliest date of occurrence of any of the events necessary for each different UTI definition. For example, the definition of UTI in our primary analysis is a combination of an UTI-related diagnostic code and antibiotic prescription in WLGP data and a urine culture result that supports a diagnosis of UTI in the WRRS data, occurring within a 7-day window. In this case, the date of the UTI would be the date of whichever of the following events occurred first: UTI-related diagnostic code, antibiotic prescription, supporting urine culture.

Individuals can have more than one UTI during the observation period, and therefore can have more than one 90-day risk period. Where risk periods overlap, the later period takes precedence, and the earlier period is shortened. There will be a prerisk period of 7 days before the risk period, to allow for the situation where an individual has an UTI for several days prior to consultation, so

events in the prerisk period are not erroneously attributed to the baseline period.

Baseline periods are all times outside of the risk and prerisk periods. The study period is from 1 January 2010 to 31 December 2020. The observation period is different for each individual. It generally follows the study period but may start later for some individuals who were not Welsh residents at the start of the study period but became so later, or who turned 30 years of age sometime during the study period. Similarly, the observation period may end before the study period when individuals moved out of Wales or died.

## Population

The SCCS method starts with identifying individuals who have had the outcome of interest. Therefore, the source population are individuals within the SAIL Databank who have a hospital admission record for acute MI or stroke during the study period. For inclusion, individuals will need to be aged 30–100 between 1 January 2010 and 31 December 2020 and be Welsh residents. We chose a lower age bound of 30 years to reduce the chance of including MIs and strokes due to congenital or other non-artherosclerotic causes. A lower age bound of 40 might reduce this chance further but would increase the chance of missing relevant events, especially given the greater burden of cardiovascular disease in Wales compared with the UK as a whole.[35]

## Outcomes

Outcomes of interest are acute MI or stroke, as identified by ICD-10 codes from inpatient diagnoses recorded in the PEDW. A list of ICD-10 codes to be used are given in online supplemental material A.

## Exposure

The exposure of interest is an UTI. The risk period is 0–90 days following an UTI. This period was chosen as previous research has shown an increase in acute MI and stroke risk in the 1–90 days following an UTI.[6] There will be a prerisk period of 7 days before the risk period, to allow for the situation where an individual has an UTI for several days prior to consultation, so events in this period are not erroneously attributed to the baseline period. Individuals can be exposed to an UTI more than once during the observation period. Each exposure will be followed by the same 90-day risk period. Baseline periods are all other times.

To ascertain UTI, we developed definitions that reflected the Public Health Wales Microbiology Division's standard operating procedure for the investigation of urine.[36] These procedures are followed by NHS microbiology laboratories across Wales. For each definition, the data sources required and the clinical scenario represented is summarised in table 2, and the code lists used are given in online supplemental material B–D. In our primary analysis, an individual will be regarded as being

**Table 3** Characteristics of cases of stroke and MI

| Characteristic | Stroke | MI |
|---|---|---|
| Sex (% male) | 49.4 | 62.8 |
| Age of males (mean (25th, 75th centiles)) | 74 (64, 82) | 69 (59, 78) |
| Age of females (mean (25th, 75th centiles)) | 79 (69, 87) | 77 (66, 85) |
| Welsh Index of Multiple Deprivation (WIMD) 2019 quintiles* | | |
| 1 (least deprived) | 19.8 | 20.6 |
| 2 | 20.8 | 20.5 |
| 3 | 20.3 | 20.9 |
| 4 | 20.4 | 20.1 |
| 5 (most deprived) | 18.6 | 17.9 |
| Current smoker (%) | 19.8 | 24.2 |
| electronic Frailty Index (mean (SD)) | 0.16 (0.12) | 0.15 (0.12) |
| Prescribed lipid-lowering drugs (%) | 49.0 | 49.6 |
| Prescribed aspirin (%) | 43.7 | 41.3 |
| Prescribed hypertensive drugs (%) | 59.1 | 55.1 |
| Prescribed beta blockers (%) | 39.6 | 37.3 |
| Chronic kidney disease (%) | 20.9 | 18.9 |
| COPD (%) | 9.9 | 10.9 |
| Asthma (%) | 15.1 | 16.0 |
| Hypertension (%) | 51.2 | 46.2 |
| Diabetes (%) | 19.4 | 20.7 |
| Cardiovascular disease (%) | 56.4 | 53.4 |
| Coronary Heart Disease (%) | 15.9 | 21.5 |
| Atrial fibrillation (%) | 16.0 | 8.7 |
| Heart failure (%) | 8.4 | 8.6 |
| Peripheral vascular disease (%) | 7.2 | 7.7 |
| Angina (%) | 11.3 | 16.0 |
| Transient ischaemic attacks (%) | 9.4 | 5.0 |
| Total no of cases | 58 150 | 51 659 |

*LSOA version 2011 and WIMD version 2019
COPD, Chronic Obstructive Pulmonary Disease; LSOA, Lower Super Output Area; MI, myocardial infarction.

exposed to an UTI if the following events occur within a 7-day window:

1. A General Practice (GP) record of an UTI diagnostic or symptom code.
2. A GP record of an antibiotic prescription.
3. A microbiology record of a urine sample with bacterial growth of a single organism of $\geq 1 \times 10^8$ colony-forming units (CFU) per litre and white cell count (WCC)$\geq 1 \times 10^8$/L. If there are two organisms grown, both must demonstrate growth of $\geq 1 \times 10^8$ CFU per litre. More than two organisms will be regarded as mixed growth and thus not supportive of an UTI diagnosis. In a sensitivity analysis, we will widen the microbiological criteria and include all urine samples with bacterial growth of a single organism of $\geq 1 \times 10^7$ CFU per litre, irrespective of the WCC.

In secondary analysis 1, we will estimate the risk of MI and stroke among individuals with a GP record of an UTI code and antibiotic prescription, and a microbiology record of a urine sample with mixed bacterial growth (any descriptor for 'mixed growth' or >3 organisms). This is an important analysis given the uncertain clinical significance of mixed bacterial growth in an individual with symptoms of UTI. In secondary analysis 2, an individual will be regarded as exposed to UTI with only a GP record of a diagnostic or symptom code and an antibiotic prescription (no microbiology). Secondary analysis 3 will estimate the risk of MI and stroke among individuals where UTI was suspected and treated by the GP, but urine microbiology showed bacterial growth of$<1 \times 10^7$CFU per litre (not supportive of an UTI diagnosis). Secondary analysis 4 will focus on individuals with a hospital admission with an UTI-related ICD-10 code and a microbiology record of a urine sample with bacterial growth of a single organism of $\geq 1 \times 10^8$ CFU per litre and white blood cells $\geq 1 \times 10^8$ per litre. As for the primary analysis, if there are two organisms grown, both must demonstrate growth of $\geq 1 \times 10^8$ CFU per litre, and more than two organisms will be regarded as mixed growth. Secondary analysis 5 combines the primary analysis, and secondary analysis 4, considering individuals with either a GP record of an UTI code and antibiotic prescription, or a hospital admission with an UTI-related ICD-10 code, and a microbiology record of a urine sample with bacterial growth of a single organism of $\geq 1 \times 10^8$ CFU per litre and white blood cells $\geq 1 \times 10^8$ per litre. In the primary analysis, any hospital diagnosed UTI is likely to count towards baseline time, whereas they would be included in exposed time here.

### Statistical analysis

We will describe the demographics of the study population, such as age and sex distribution, medical history and prescribed medication, using means and SDs for continuous variables, and frequencies and proportions for categorical variables. For each analysis, we will use conditional logistic regression to estimate IRRs, with 95% CIs, for the risk of acute MI or stroke in prerisk and risk periods compared with baseline periods. We will include only individuals who have experienced both the outcome and the exposure, and will include only the first acute MI or stroke diagnosis in the observation period. The IRRs will be adjusted for time-varying confounders: age, season and year of UTI diagnosis. Year of UTI diagnosis is included because diagnostic and coding practices may have changed over time as a result of guidance and awareness around microbial resistance. Adjusted IRRs will be reported for the risk of acute MI or stroke at 1–7, 8–14, 15–28 and 29–90 days after UTI. We will conduct the analysis in R, using the SCCS package. We will report the findings in accordance with REporting of studies Conducted using Observational Routinely-collected Data (RECORD) and STrengthening the Reporting of OBservational studies in Epidemiology (STROBE) guidelines.

### Sensitivity and subgroup analyses

We will perform several sensitivity and subgroup analyses to assess the robustness of the findings of our primary analysis to different assumptions:

- ► We will explore the impact of using a wider definition of MI and stroke, including ICD-10 codes for acute coronary syndromes and transient ischaemic attacks. This will include events that have potentially been missed by our main definition and assess how sensitive our findings are to the definition of MI and stroke.
- ► We will explore the impact of widening the microbiological definition of UTI to bacterial growth of a single organism of $\geq 1 \times 10^7$ CFU per litre irrespective of WCC.
- ► We will differentiate first-ever MI or stroke from recurrent events, and report risk estimates separately. This analysis will exclude individuals with a PEDW record of an event before the observation period, and include only those who have their first ever event during the observation period.
- ► We will extend the prerisk period to 14 days. Some individuals may have had an UTI for several days prior to diagnosis, and so an acute MI or stroke during this time may relate to exposure but without a prerisk period, would count towards the baseline period.
- ► We will repeat the analysis excluding individuals who died within 30 days of an event to examine the potential effect of an event dependent observation period.
- ► We will restrict the definition of UTI to include only antibiotic prescriptions for nitrofurantoin (currently recommended first-line therapy) to explore whether the choice of antibiotic impacts the findings.
- ► We will use interaction terms to assess the impact of specific bacterial organisms on the relationship between UTI and MI or stroke.
- ► We will examine whether the COVID-19 pandemic may affect our findings by (1) excluding individuals whose MI or stroke occurred in 2020, and (2) including an interaction term to explore whether the association between exposure and outcome differs in 2020 vs pre-2020.
- ► We will include an interaction term to explore whether the association between exposure and outcome differs in those with and without a history of diabetes, given its potential role as a risk factor of both UTI and MI/stroke.

### Sample size and power

Our initial work has identified 51 656 individuals with acute MI and 58 146 with stroke in the SAIL Databank who meet all inclusion criteria. In the previous study by Smeeth *et al*,[6] the sample size was 53 709 for acute MI, and 55 157 for stroke, where 16% of acute MI cases, and 21% of stroke cases were exposed to UTI.[6] Based on a conservative exposure rate of 10%, and an at-risk window of 90 days, we estimated the effect size that could be reliably detected with our potential sample size, using the sample size function in the SCCS package in R. The available sample provides 90% power to detect an IRR of 1.3 at the alpha=0.05 level, which is smaller than the IRRs detected in Smeeth *et al*[6]

### Patient and public involvement

We developed this research proposal in collaboration with members of the Wales Centre for Primary and Emergency Care Research Service Users group (SUPER). We have a patient and public involvement (PPI) plan and are consulting the SAIL consumer panel and SUPER regarding all stages of this research, including ongoing discussion of analysis plans, review of findings, and plans for dissemination (eg, public facing outputs). We have extended our sub-group analysis to include individuals with diabetes in response to discussions with the SAIL consumer panel members. Identifying PPI for future stages of this research is a secondary objective of our PPI plan.

### Summary of cases

The individuals with an ICD-10 code for either MI or stroke were selected according to the eligibility criteria, as shown in figure 2. There are 58 146 individuals with an ICD-10 code for stroke, and 51 656 individuals with an ICD-10 code for MI, a total of 105 930 unique individuals (it is possible for an individual to be in both the stroke and the MI group). Stroke cases were 49% male, MI cases 63% male. Female stroke cases were older than males, with a median age of 79 years (25–75th centiles 69–87) compared with a median of 74 years (25–75th centiles 64–82) for males. Female MI cases were also older, with a median age of 77 (25–75th centiles 66–85) compared with 69 (25–75th centiles 59–78) for males.

The mean length of observation (study entry to study exit) of stroke cases was 2796 days for females and 2981 days for males. For MI cases, 2981 for females and 3251 for males. History of diagnoses and prescription drugs is taken from all health records available for each individual. Health data is available for a median of 13 years (25–75th centiles 8–16) prior to the study start.

Characteristics of cases, including a history of diagnoses and prescription drugs prior to an event, smoking status, Welsh Index of Multiple Deprivation version 2019 and electronic Frailty Index are given in table 3.

### Ethics and dissemination

All study data will be held within the SAIL Databank, an ISO27001 certified TRE for anonymised individual-level data. Data access, research permissions and approvals have been obtained from the SAIL independent IGRP, project number 0972. Analyses will be conducted within the SAIL TRE. There are strict disclosure control processes in place. Only aggregated outputs will be approved for release to ensure individuals are not identified. Findings will be disseminated through peer-review journals and conferences. Results will be of interest internationally to primary and secondary care clinicians who manage UTIs. An association between UTI and either MI or stroke will support a future funding application for a randomised trial of preventative treatments, such as anti-platelet drugs.

**Author affiliations**
[1]Division of Population Medicine, Cardiff University, Cardiff, UK
[2]Population Data Science, Swansea University Medical School, Swansea, UK
[3]Centre for Trials Research, Cardiff University, Cardiff, UK
[4]PRIME Centre Wales, Division of Population Medicine, Cardiff University, Cardiff, UK
[5]Specialist Antimicrobial Chemotherapy Unit, Public Health Wales, University Hospital of Wales Healthcare NHS Trust, Cardiff, UK

**Acknowledgements** This study makes use of anonymised data held in the Secure Anonymised Information Linkage (SAIL) Databank. We would like to acknowledge all the data providers who make anonymised data available for research.

**Contributors** HA is the chief investigator of the study. All authors have contributed to and are responsible for the final design of the study. NFR is responsible for study management. NFR, VB, FT and AA are responsible for the data management. NFR and RC-J are responsible for statistical planning and analysis. FVL-W is leading the patient and public involvement plans for the study. HA and MW are responsible for the microbiological definition of UTI. All authors have read and approved the final manuscript (NFR, VB, DG, KH, FVL-W, RC-J, FT, MW, AA and HA).

**Funding** This work was supported by The British Heart Foundation, grant number PG/20/10419. The Centre for Trials Research is funded by Health and Care Research Wales (grant number N/A) and Cancer Research UK (grant number N/A). Population Data Science, Swansea University is supported by Health Data Research UK (HDR-9006) and ADR Wales (grant ES/S007393/1). PRIME Centre Wales is funded by Health and Care Research Wales.

**Competing interests** None declared.

**Patient and public involvement** Patients and/or the public were involved in the design, or conduct, or reporting, or dissemination plans of this research. Refer to the Methods section for further details.

**Patient consent for publication** Not applicable.

**Provenance and peer review** Not commissioned; externally peer reviewed.

**ORCID iDs**
Nicola F Reeve http://orcid.org/0000-0001-9602-6675
David Gillespie http://orcid.org/0000-0002-6934-2928
Fiona V Lugg-Widger http://orcid.org/0000-0003-0029-9703
Rebecca Cannings-John http://orcid.org/0000-0001-5235-6517
Ashley Akbari http://orcid.org/0000-0003-0814-0801

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
