## [Reviewer comments · BMJ Open]

ARTICLE DETAILS

TITLE (PROVISIONAL)	Myocardial Infarction and stroke subsequent to urinary tract infection (MISSOURI): protocol for a self-controlled case series using linked electronic health records
AUTHORS	Reeve, Nicola F; Best, Victoria; Gillespie, David; Hughes, Kathryn; Lugg-Widger, Fiona; Cannings-John, Rebecca; Torabi, Fatemeh; Wootton, Mandy; Akbari, Ashley; Ahmed, Haroon

VERSION 1 – REVIEW

REVIEWER	Kaier , Thomas King's College London, Rayne Institute
REVIEW RETURNED	06-Jul-2022

GENERAL COMMENTS	The study protocol is sound and I have no additional comments.
--

REVIEWER	Lee, Joseph University of Oxford, Nuffield Department of Primary Care Health Sciences
REVIEW RETURNED	26-Jul-2022

GENERAL COMMENTS	Myocardial Infarction and stroke subsequent to urinary tract infection (MISSOURI): protocol for a self-controlled case series using linked electronic health records To my mind this is an important question, with strong clinical relevance, and I suspect we are going to be glad to know the answer, if positive or negative. I agree this study design is an appropriate way to investigate the causal question, and I'm glad you did not shy away from using causal language. First thoughts on objectives: – why 90 days (I see this is mentioned in the Exposure section)? My simple minded approach to the mechanisms between infection and CVD are that 1. it might be accelerating atherosclerosis, or 2. it might be triggering existing disease (by increased cardiac output, type ii MI, pro-thrombotic states, etc etc), but with either of these I imagine the risk is likely highest in the early days. ? room for sensitivity analyses or multiple analyses as was done in the earlier RTI work? I suppose I'm saying I think it is interesting to know the period of risk as were we to be intervening in some way, this is going to be important information to know who is at risk and how long one
--

	should intervene for.  - Is ascertainment really a proxy for severity? - Organism is an interesting idea - What about antibiotics? Is there a natural experiment here? – are people started on antibiotics who are then found to have a resistant organism at higher risk? I appreciate table 1 – it is very helpful. Population – The age – why start at 30? Pretty rare to have an MI or stroke in the 30s and are probably going to be due to slightly unusual causes, such as central venous thrombosis, trauma, subarachnoids, cancers, anomalous anatomy etc. As you are using first events I suspect a higher lower age bound would increase the relationship you are looking for, even though the numbers might be lower. Outcomes – any distinction between subtypes of stroke? (Though I appreciate this is horribly coded, in CPRD data at least - see Davidson for grim details). But to clarify, is this including TIA? Spinal strokes? Subdurals? (UTI, falls over, subdural is probably not the sequence of events that is of interest). Exposure – See above – I think an analysis with different time periods would be interesting, shorter ones in particular. Overall – very interesting study.
--	---

VERSION 1 – AUTHOR RESPONSE

Reviewer 1:

We thank the Reviewer for his comments. There are no issues for us to address.

Reviewer 2:

Comment: To my mind this is an important question, with strong clinical relevance, and I suspect we are going to be glad to know the answer, if positive or negative. I agree this study design is an appropriate way to investigate the causal question, and I'm glad you did not shy away from using causal language.

Response: Thank you for the positive feedback about our study.

Comment: First thoughts on objectives:

– why 90 days (I see this is mentioned in the Exposure section)? My simple minded approach to the mechanisms between infection and CVD are that 1. it might be accelerating atherosclerosis, or 2. It might be triggering existing disease (by increased cardiac output, type ii MI, pro-thrombotic states, etc etc), but with either of these I imagine the risk is likely highest in the early days. ? room for sensitivity analyses or multiple analyses as was done in the earlier RTI work? I suppose I'm saying I think it is interesting to know the period of risk as were we to be intervening in some way, this is going to be important information to know who is at risk and how long one should intervene for.

Response: We agree with the Reviewer that time period of greatest risk is important to determine. This is why we will estimate Incidence rate ratios (IRR) for 1-7, 8-14, 15-28 and 29-90 days after UTI for both MI and stroke (as stated in section 2.7 Statistical Analysis, page 10). If we find an effect of

UTI, we hypothesise that the risk will be highest in the early days as you suggest, then decline with time. This would be reflected in higher IRRs for the earliest periods, and lower IRRs for the later periods.

Comment: Is ascertainment really a proxy for severity?

Response: The different definitions of UTI in the primary and secondary analyses provide ascertainment in different clinical scenarios. These clinical scenarios are a proxy for severity: a clinician not requesting a urine sample would usually indicate milder cases. A clinician requesting a urine sample would often indicate greater severity of symptoms or that a treatment has failed. UTIs diagnosed in hospital would usually be more severe cases.

Comment: Organism is an interesting idea

Response: We agree – we plan to do sub-group analyses to assess the impact of different organisms. We have clarified this in the Sensitivity and sub-group analyses section on page 11.

Comment: What about antibiotics? Is there a natural experiment here? – are people started on antibiotics who are then found to have a resistant organism at higher risk?

Response: This is an interesting question. We will conduct a sensitivity analysis restricting the definition of UTI to include only antibiotic prescriptions for nitrofurantoin (currently recommended 1st line therapy) to explore the effect of the choice of antibiotic, as stated in section 2.8, Sensitivity and sub-group analyses, page 11. As stated above, we will also examine the impact of different organisms. We are not examining the effect of resistant organisms as this is outside the scope of the main study, but we may include this as a sub-study if time and budget allows.

Comment: Population – The age – why start at 30? Pretty rare to have an MI or stroke in the 30s and are probably going to be due to slightly unusual causes, such as central venous thrombosis, trauma, subarachnoids, cancers, anomalous anatomy etc. As you are using first events I suspect a higher lower age bound would increase the relationship you are looking for, even though the numbers might be lower.

Response: We chose a lower age bound of 30 years to reduce the chance of including MIs and strokes due to congenital or other non-atherosclerotic causes. A lower age bound of 40 might reduce this chance further but would increase the chance of missing relevant events, especially given the greater burden of cardiovascular disease in Wales compared to the UK as a whole. Wales has higher years of life lost and disability-adjusted life-years than England [[https://www.thelancet.com/journals/lancet/article/PIIS0140-6736\(18\)32207-4/fulltext](https://www.thelancet.com/journals/lancet/article/PIIS0140-6736(18)32207-4/fulltext)], suggesting a lower age of onset of cardiovascular disease.

Comment: Outcomes – any distinction between subtypes of stroke? (Though I appreciate this is horribly coded, in CPRD data at least - see Davidson for grim details). But to clarify, is this including TIA? Spinal strokes? Subdurals? (UTI, falls over, subdural is probably not the sequence of events that is of interest).

Response: We are not planning to distinguish between different subtypes of strokes due to the coding issues alluded to by the Reviewer. TIAs will be included in a sensitivity analyses that uses a wider definition of MI and stroke. This is described in section 2.8 Sensitivity and sub-group analyses, page 11. We are not including subdural haemorrhage or extradural haemorrhage for exactly the reason the reviewer suggests.

Comment: Exposure – See above – I think an analysis with different time periods would be interesting, shorter ones in particular.

Response: See response above -incidence rate ratios (IRR) will be reported for 1-7, 8-14, 15-28 and 29-90 days.

Comment: Overall – very interesting study.

Response: We thank the reviewer for their kind comments.

VERSION 2 – REVIEW

REVIEWER	Lee, Joseph University of Oxford, Nuffield Department of Primary Care Health Sciences
REVIEW RETURNED	17-Aug-2022
GENERAL COMMENTS	This will be an interesting and important study.